# Receiving quality antenatal care service increases the chance of maternal use of skilled birth attendants in Ethiopia: Using a longitudinal panel survey

**Shikur Mohammed**[1,2]*, **Alemayehu Worku**[2], **Eshetu Girma**[2]

**1** School of Public Health, Saint Paul's Hospital Millennium Medical College, Addis Ababa, Ethiopia, **2** School of Public Health, College of Health Sciences, Addis Ababa University, Addis Ababa, Ethiopia

* shikurmohamed@yahoo.com

## Abstract

### Background

Evidence has suggested that maternal use of skilled birth attendant is the most important factor to reduce maternal mortality because of more than three-fourths of maternal deaths occur during child delivery or within 24 hours after delivery due to hemorrhage, hypertension, ruptured uterus and sepsis. In Ethiopia, more than 42% of pregnant women with 4+ antenatal care (ANC) visit did not deliver by skilled birth attendant. The factors for women not using skilled birth attendant after any ANC visit is not well-studied yet. Therefore, the aim of this study was to assess effect of quality antenatal care service on maternal use of skilled birth attendant after any antenatal care visit.

### Methods

This study was initiated using performance monitoring for action Ethiopia longitudinal panel survey datasets. A total of 1,511 postnatal women were included in the analysis. Generalized estimating equation Poisson regression model was used to assess the effect of quality ANC service on maternal use of skilled birth attendant by accounting the clustering nature of the data using Stata vers.16 software.

### Results

This study showed that about 54% of women used skilled birth attendant during the current baby delivery (rate = 53.6%, 95% UI = 51–56%). Nearly, 39% of the women received quality (more than 75th percentile) ANC service (rate = 39.05%, 95% UI = 36–42%). The highest and the lowest antenatal care service the women received, respectively, were blood pressure measure (91.9%) and syphilis test (12.4%). Women who received better quality ANC service were 20% higher more likely to use skilled birth attendant compared to women who received low quality ANC service (adjusted IRR = 1.20, 95% UI = 1.11, 1.31).

**Data Availability Statement:** Data cannot be shared publicly because of we ourselves accessed from performance monitoring for action Ethiopia project. Data are available from the PMA-Ethiopia

website (contact via info@pmadata.org) for researchers who meet the criteria for access to confidential data.

**Funding:** The authors received no specific funding for this work.

**Competing interests:** The authors have declared that no competing interests exist.

## Conclusion

Maternal use of skilled birth attendant can be improved by providing quality ANC service during subsequent ANC visits. Improving ANC service delivery may encourage or positively reinforce women's and partner's decision to use skilled birth attendant.

## Introduction

World Health Organization (WHO) and United Nations Children Funds (UNICEF) strongly advised the global community to accelerate efforts to reach the global Sustainable Development Goals (SDG) targets of less than 70 maternal deaths per 100, 000 live births by 2030 [1, 2]. However, the progress to achieve this target is slow in most developing countries including Sub-Saharan African countries, where nearly 73% of maternal deaths occurred in these countries due to direct obstetric causes. More than three-fourths of those deaths occurred during child delivery or within 24 hours after delivery due to hemorrhage, hypertension, ruptured uterus and sepsis [2–6], yet majority of those deaths could have been prevented if women had used skilled birth attendant [7].

Different factors have influenced to ensure women deliver in health facilities by a skilled birth attendant [8]. The underlying barriers of maternal use of skilled birth attendant can be categorized as predisposing factors (sociodemographic and set of beliefs towards behaviour), enabling factors (economic or transport or service access barriers), and quality of maternal healthcare services [9]. Several studies were conducted to explore factors influencing maternal use of skilled birth attendant, but the reasons for maternal not use of skilled birth attendant particularly after women had any antenatal care (ANC) visit is not well studied. Few studies have reported that the odds of maternal use of skilled birth attendant after any ANC visit was higher among women who had received main ANC services, which include blood pressure measurement, blood and urine sample examination, vaccination for tetanus, information on danger signs, and message on nutrition and iron tablets [10–13]. In addition, studies have reported that the odds of maternal use of skilled birth attendant increased by 30 to 50% if women had blood pressure measurement, and gave urine and blood samples as part of antenatal care services [11, 14].

Evidence has shown that antenatal care is the key means to identify and prevent progression of the early onset of predominant causes of maternal and perinatal morbidity and mortality in low-and middle–income countries including Ethiopia: the predominant causes of maternal mortality in these countries are postnatal hemorrhage, preeclampsia/eclampsia and infection [15]. As a result of this, WHO envisions a world where every pregnant woman and newborn receives quality care along the continuum of care due to pregnancy-related complication and mortality remain unacceptably high. ANC provides an opportunity to identify any pre-existing women's medical, obstetrical and social factors to monitor weight gain and assess nutritional status, and to identify those risk factors arising during current pregnancy and to ensure a safe transition to positive labor and birth [10, 15, 16]. However, the effect of quality ANC service on maternal use of skilled birth attendant is not well studied.

In Ethiopia maternal use of skilled birth attendant improved from 28% in 2016 to 50% in 2019; however, more than 42% of pregnant women with four or more ANC visit did not deliver by skilled birth attendant [3, 17]. The factors for women not using skilled birth attendant after any number of ANC visit is not well-studied yet [3, 4, 10]. Therefore, the objective of this study was to measure the effect of quality ANC service on maternal use of skilled birth

attendant after having at least one ANC visit to improve future maternal use of skilled birth attendant or facility delivery.

## Materials and methods

### Study settings and design

This study was initiated using performance monitoring for action Ethiopia (PMA-Ethiopia) longitudinal panel survey datasets. Performance monitoring for action -Ethiopia survey was conducted in six regions that collectively represent 90% of the population in Ethiopia, namely, Addis Ababa, Oromia, Amhara, Tigray, former Southern Nations Nationality People (SNNP) and Afar regions. The details of performance monitoring for action Ethiopia project settings can be accessed from the published PMA-Ethiopia project protocol [18].

In PMA-Ethiopia, longitudinal panel survey design was used to identify service delivery gaps in relation to maternal and newborn healthcare. In this panel survey, women were screened for pregnancy (self-identified as current pregnant) and enrolled, regardless of their gestational age, from 217 enumeration areas (EAs) in six regions of the country. Then, women expected date of delivery was estimated at the time of enrollment and followed after delivery at 6-weeks of postnatal period to understand the trends of health seeking behavior, utilization and quality. Therefore, this study was initiated using the PMA-Ethiopia 2019–21 datasets to assess influence of quality ANC services (process) on maternal use of skilled birth attendant.

### Study population and sampling techniques

All postnatal women who were enrolled during pregnancy and received ANC service from skilled attendants from selected enumeration areas in the six regions were included. The required sample size for this study was calculated using Stata vers.16 software considering the following assumptions: a 10% outcome difference between exposed and unexposed, a 42% baseline level of use of skilled delivery in women who received quality ANC service (key indicators) chosen from studies [19], 5% level of significance, and 80% power. The required sample size was increased by design effect (Deff = 1.5) [20] and non-response or incompleteness rates (10%). Hence, the estimated sample becomes 1, 098. The PMA- Ethiopia panel survey employed stratified two-stage cluster sampling design to select enumeration areas (clusters) and households from the six regions. In the first stage, each region was stratified into urban and rural, then enumeration areas (EAs) were selected from both strata using central statistical agency (CSA) sampling frame, which contains information about the EAs location, type of residence (urban, rural) and the estimated number of households in each EAs. Clusters were selected using probability proportional to size sampling technique. At the second stage, 35 fixed number of households were selected from each cluster and all pregnant women in a given household were enrolled and followed. In this study, to increase the power of the study, all available 1,511 women who had any number of ANC visits during their pregnancy were included in the final analysis.

### Measurement, data management and analysis

In PMA-Ethiopia panel survey, a structured and standardized enrolment and follow up forms were used for interviewing women to get information about maternal, newborn and infant related healthcare. Data were collected by using Open Data Kit (ODK) system using tablet computers by trained field workers. The baseline sociodemographic and reproductive details of the women including women age, women age at first birth, women educational and employment status, women economic decision power, area of residence, household wealth quintile,

husband education and occupation, ethnicity, religion/province, family size, parity and pregnancy desire, husband and community encouragement to use maternal and child healthcare services were collected at the time of enrollment. Similarly, Pregnancy related data on receipt of ANC services (ANC components), pregnancy complications (antepartum hemorrhage, diabetes mellitus, preeclampsia), infection (malaria, HIV, syphilis, Hepatitis-B), anemia, labor and delivery related data on mode of delivery, duration of labor, type of birth attendant, place of delivery and presence of birth complication were collected during follow up interview.

**Skilled delivery** is defined as women receipt of skilled assistance during delivery by skilled birth attendant. Skilled birth attendant is defined as "Registered (accredited) midwife or a health worker with midwifery skills (Nurse, medical doctor and health officer) who has been educated and trained to proficiency in the skills to manage normal (uncomplicated) pregnancies, childbirth, and the immediate postnatal period, and in the identification, management and referral of complications in women and newborns [21, 22]. To determine skilled birth assistance, women were asked a question of "did you deliver by a skilled attendant?", and asked a follow-up question of "type of professional skilled birth attendant."

**Antenatal care service.** Quality of ANC service is defined as women receipt of ANC content packages, which includes blood pressure and weight measurement, give blood, urine and stool samples during ANC visit, tetanus toxoid injection, received nutrition information during ANC visit *(eat more (quantity)*, *eat variety/iron rich food*, *take iron tablets*, *take malaria preventive treatment*, *take deworming tablet*, *how much weight gain*, *exercise regularly*, *how to manage nausea/vomiting*, *reduce salt intake*, *do not eat raw meat)*, discussed transport for delivery during ANC, discussed place of delivery during ANC visit, discussed delivery by skilled birth attendant during ANC visit, discussed place to go if experience danger signs *(headaches*, *blurred vision*, *high blood pressure*, *edema*, *convulsion/fits*, *bleeding)*, HIV tested as part of ANC and Syphilis tested as part of ANC [10, 14, 23]. The ANC service quality was determined by giving scores to these 13 ANC services the women received during pregnancy. The reliability of these ANC services was checked and the reliability coefficient is greater than 0.7 (alpha = 0.756). The score ranged from 1 to 13, and the value 11 gave a 75th percentiles. Then, women were classified as received better quality ANC service if they received more than the 75th percentiles of ANC service. The quality score was divided into above '75th percentile' and below '75th percentile', and reported as 'better' and 'low' quality ANC service, respectively [19, 24].

Household wealth was determined by giving scores based on number and kinds of consumer goods they own, source of drinking water, type of toilet facilities and flooring materials. Scores was derived using principal component analysis, and then households' wealth index was divided into quintiles according to the wealth score as "lowest", "lower", "middle", "higher" and "highest" [4].

Data were formally requested and downloaded from the PMA Ethiopia website [25, 26]. Data were cleaned and prepared for merging using Stata vers.16 software. The baseline and the six-week postnatal visit data were merged using the study participant identity number (participant ID). Then, the appropriate sample was restricted, and ANC service quality score was created and transformed into categorical exposure variable. Description of the study participants, unweighted and weighted frequency, by sociodemographic characteristics was done and presented using table. Sampling weight (the inverse of a sample selection probability) was applied to account the non-equal selection probability of the individuals and improve representativeness.

Different models were fitted to estimate the effect of quality ANC service on maternal use of skilled birth attendant. First, unweighted and weighted poison regression models with robust variance were fitted to account effect of the complex sampling design used by the study.

Then, the mean square error (MSE) value of each model was computed using the formula *(MSE = Bias² + StdErr² where, bias = (weighted estimate–unweighted estimate)* and compared. The MSE of the unweighted poison regression model was smaller than the MSE of the weighted poison regression model. Therefore, not accounting the use of complex sampling design in the model fitting will not affect the estimates.

Then, as we have a two-level cluster data (correlated data), generalized estimating equation (GEE) Poisson regression model was fitted to account the correlation of observations within an enumeration area (cluster). The full GEE poison regression model was fitted by adjusting the effect of a potential confounder variables including area of residence, partner encourage for facility delivery, child desire before pregnancy and number of ANC visits a women had. These variables were selected as a potential confounding variable based on a p-value cutoff of less than or equal to 0.05 during bi-variable analysis. Incidence rate ratio (IRR) with 95% confidence interval (CI) was computed to measure the effect of the exposure variable on maternal use of skilled birth attendant and interpreted.

The study was initiated after getting the ethical clearance permission from Addis Ababa University College of Health Sciences. The PMA-Ethiopia project got its ethical clearance and all the necessary approvals and permission from Addis Ababa and John Hopkin universities research ethics committee, and other concerned stakeholders (regional health bureau's). Therefore, the authors formally requested access to the necessary PMA datasets with the appropriate data dictionary. All data were fully anonymized before we accessed them. The researchers have respected and obeyed the PMA-Ethiopia data use and share policy by not sharing to third person who is outside the current research team. Data received from PMA Ethiopia were password protected.

## Results

### Sociodemographic characteristics of antenatal care attendant women

In this study a total of 1, 511 women who had at least one number of ANC visit during their pregnancy were included from six regions of Ethiopia. About 42% of the women were from Oromia region, followed by Amhara (25.3%) and SNNP (19.6%) regions. Nearly three-forth (73.7%) of the study participant women were from the rural part of the regions.

Higher proportions (33.1%) of women were in age categories between 25–29 years and followed by about 23% were in age categories between 20–24 years old. Nearly forty-two percent of the study participant women were Orthodox Christian by religion and followed by Muslim (34.7%). Nearly 17% of the women had no desire for children before they became pregnant. More than 53% of women in the study settings had first time ANC visit at more than 4 months of their pregnancy. **(See Table 1).**

### Variables for Antenatal care quality index

Women were classified as received better quality ANC service if they received more than the 75th percentiles of ANC service. Therefore, based on this classification only 39% of women received better quality ANC service. **(See Table 2)**

### Effect of quality ANC service on maternal use of skilled birth attendant after any ANC visit

This study showed that about 54% of the women used skilled birth attendant during their current baby delivery. The rate of maternal use of skilled birth attendant was 20% higher among

**Table 1. Sociodemographic characteristics of antenatal care attendant women in six regions of Ethiopia (n = 1,511), 2019–21.**

| Sociodemographic characteristics | Weighted percent | Weighted N | Unweighted N |
|---|---|---|---|
| **Region** | | | |
| Tigray | 8.4 | 127 | 321 |
| Afar | 0.4 | 6 | 27 |
| Amhara | 25.3 | 382 | 322 |
| Oromia | 42.3 | 639 | 377 |
| SNNP | 19.6 | 296 | 307 |
| Addis Ababa | 4.0 | 61 | 157 |
| **Marital status** | | | |
| Married/in union | 97.7 | 1,477 | 1, 467 |
| Single | 2.3 | 34 | 44 |
| **Women Age (in years)** | | | |
| 15–49 | 9.9 | 150 | 122 |
| 20–24 | 23.1 | 349 | 364 |
| 25–29 | 33.1 | 500 | 521 |
| 30–34 | 18.4 | 278 | 276 |
| > = 35 | 15.5 | 234 | 228 |
| **Religion** | | | |
| Orthodox | 41.9 | 633 | 806 |
| Islam | 34.7 | 525 | 390 |
| Protestant | 21.3 | 322 | 295 |
| Other* | 2.1 | 31 | 20 |
| **Area of residence** | | | |
| Urban | 26.3 | 397 | 688 |
| Rural | 73.7 | 1,114 | 823 |
| **Household wealth index** | | | |
| Lowest quintile | 17.1 | 258 | 207 |
| Lower quintile | 16.5 | 250 | 178 |
| Middle quintile | 20.6 | 312 | 245 |
| Higher quintile | 21.6 | 326 | 283 |
| Highest quintile | 24.2 | 365 | 618 |
| **Desire for children before became pregnant** | | | |
| Wanted to have a baby | 73.8 | 1,115 | 1,175 |
| Had mixed feelings about to have a baby | 9.0 | 136 | 133 |
| Did not want to have a baby | 17.2 | 260 | 203 |
| **Time of first ANC initiation** | | | |
| < = 4 months | 47.0 | 710 | 825 |
| >4 months | 53.0 | 801 | 686 |
| **Number of times women received ANC from health care provider** | | | |
| Less than 4 ANC visits | 56.4 | 853 | 699 |
| > = 4 ANC visits | 43.6 | 658 | 812 |

*Catholic, Wakefta, traditional

women with better quality ANC service compared to women who received low quality ANC service (adjusted IRR = 1.20, 95% CI = 1.11, 1.31).

In the analysis the effect of some potential confounding variables was adjusted including partner discussion where to deliver, area of residence, desire for pregnancy and number of

**Table 2. Variables for antenatal care quality index among women in Ethiopia, 2019–21.**

| Type of service received | Frequency (%) (n = 1, 511) |
|---|---|
| Blood pressure measured as part of ANC | 1, 390 (91.9) |
| Weight measured as part of ANC | 1, 343 (88.9) |
| Blood sample taken as part of ANC | 1, 314 (86.9) |
| Urine sample taken as part of ANC | 1, 017 (67.6) |
| stool sample taken as part of ANC | 532 (35.2) |
| Given tetanus injection during this pregnancy | 1, 023 (67.9) |
| Received nutrition information during ANC visit | 712 (47.1) |
| Discussed transport for delivery during ANC | 985 (65.3) |
| Discussed place of delivery during ANC visit | 1, 098 (72.7) |
| Discussed during ANC visit: delivery by skilled birth attendant | 1, 069 (70.8) |
| Discussed during ANC visit: place to go if experience danger signs | 727 (48.2) |
| HIV tested as part of ANC | 1, 161 (78.2) |
| Syphilis tested as part of ANC | 187 (12.4) |
| Scale reliability coefficient (alpha = 0.756) | |
| Received better quality ANC service | 590 (39.05) |

ANC visits. The rate of maternal use of skilled birth attendant was 32% higher among women who had discussion with their partner about place of delivery compared to women who did not have discussion with their partner about their place of delivery (adjusted IRR = 1.32, 95% CI = 1.13, 1.54). Similarly, the rate of maternal use of skilled birth attendant was higher among urban resident women and women who had four or more ANC visits during pregnancy compared to their counter parts, respectively. **(See Table 3).**

**Table 3. Effect of quality ANC service on maternal use of skilled birth attendant after any ANC visit in Ethiopia, 2019–21.**

| Variables | Use of skilled birth attendant (n = 1, 511) | | Crude IRR (95% CI) | Adjusted IRR(95% CI) |
|---|---|---|---|---|
| | Yes (%) | No (%) | | |
| **Quality of antenatal care** | | | | |
| Better | 398 (67.46) | 192 (32.54) | 1.25(1.13,1.38) | 1.20(1.11, 1.31)* |
| Low | 412 (44.73) | 509 (55.27) | 1.00 | 1.00 |
| **Discussed with partner where to deliver** | | | | |
| Yes | 736 (59.64) | 498 (40.36) | 1.39(1.20,1.59) | 1.32(1.13, 1.54)* |
| No | 74(26.71) | 203 (73.29) | 1.00 | 1.00 |
| **Area of residence** | | | | |
| Urban | 527(76.60) | 161(23.40) | 2.19(1.81,2.67) | 1.89(1.57, 2.29)* |
| Rural | 283 (34.39) | 540 (65.61) | 1.00 | 1.00 |
| **Desire for children before became pregnant** | | | | |
| Wanted to have a baby | 662 (56.34) | 513 (43.66) | 1.15(1.01,1.32) | 1.08(0.93, 1.25) |
| Mixed feelings about having a baby | 74 (55.64) | 59 (44.36) | 1.24(1.04,1.47) | 1.16(0.97, 1.39) |
| Did not want to have a baby | 74 (36.45) | 129 (63.55) | 1.00 | 1.00 |
| **Number of ANC visits** | | | | |
| > = 4 ANC visits | 554 (68.23) | 258(31.77) | 1.24(1.11,1.37) | 1.15(1.04, 1.28)* |
| < 4 ANC visits | 256(31.60) | 443(63.38) | 1.00 | 1.00 |

*statistical significance confidence interval.

## Discussion

In this study, we primarily assessed the effect of quality ANC service on maternal use of skilled birth attendant after a women had any number of ANC visit. This study revealed that women who received more than the 75th percentiles of ANC services (better quality ANC service) were more likely to use skilled birth attendant. Similarly, women living in urban setting, had 4 + ANC visit and had partner support about place of delivery were more likely to use skilled birth attendant.

Evidence has shown that the poor quality of antenatal or delivery care received by women in low-and middle income countries is contributing to the high levels of maternal mortality [27]. Similarly, evidence has suggested that maternal use of skilled birth attendant is the most important factor to reduce maternal mortality [28] because of more than three-fourths of maternal deaths occurred during child delivery or within 24 hours after delivery due to hemorrhage, hypertension, ruptured uterus and sepsis [2, 3].

The current study revealed that the rate of maternal use of skilled birth attendant was higher among women who received better quality ANC service compared to women who received low quality ANC service. This finding was consistent with previous works that revealed women who received better or adequate antenatal care services were more likely to use skilled birth attendant compared to women who received less than the 75th percentiles of antenatal care services [19, 24, 29]. This suggests that provision of quality or adequate ANC service during pregnancy had a great role in promoting skilled birth attendant or institutional delivery. Improving the quality of ANC service delivery may increase women and partner's perceived quality of care and this may lead to better utilization of facility delivery. Furthermore, provision of care may be judged of high quality against recognized standards of care but unacceptable to the women, her family and the community [27]. Therefore, attention to quality ANC service will be important to encourage or positively enforce women's and partner's decision or plan to use skilled birth attendant.

As a longitudinal panel survey, our study has many strengths. The unequal selection probability of the women and the clustering nature of the data by enumeration area were accounted in the analysis, so the findings can be reliable and used to make inference or generalization. The study also is less susceptible to recall bias as women were inquired to respond services received before few weeks. However, this study may be at risk of Hawthorne effect bias as the study women might change in using the ANC and delivery services, simply as a result of knowing being studied, but this may not be a problem in this study as the misclassification is more likely to be non-deferential in the enumeration areas. In the current study, we have assessed quality from using service process (provision) quality dimension from the women's perspective who had already at least 1st ANC visit. This is very important area, but most of the time overlooked by researchers. We haven't measure quality comprehensively from both demand and supply side. This is one of the limitations of the study.

## Conclusions

Maternal use of skilled birth attendant can be improved by providing quality ANC service during subsequent ANC visits. Attention to quality ANC service will be important to encourage or positively enforce women's and partner's decision or desire to use skilled birth attendant. Health planners and providers should assure women receipt of appropriate and respectful ANC service and educate and promote husband support in maternal use of ANC service and facility delivery by explaining the benefit of subsequent ANC service and use of skilled birth attendant for the health of women and newborn through different communications means. It is also important to further qualitatively explore barriers of quality ANC service provision and

other barriers of maternal not use of SBA especially after having skilled ANC visits to promote women health and improve use of skilled birth attendant or facility delivery.

## Acknowledgments

The authors would like to thank PMA-Ethiopia for sharing the datasets.

## Author Contributions

**Conceptualization:** Shikur Mohammed, Alemayehu Worku, Eshetu Girma.

**Data curation:** Shikur Mohammed.

**Formal analysis:** Shikur Mohammed.

**Methodology:** Shikur Mohammed, Alemayehu Worku, Eshetu Girma.

**Supervision:** Alemayehu Worku, Eshetu Girma.

**Validation:** Alemayehu Worku, Eshetu Girma.

**Writing – original draft:** Shikur Mohammed.

**Writing – review & editing:** Alemayehu Worku, Eshetu Girma.

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
