## [Decision Letter · Decision Letter 0]

28 Oct 2022

PONE-D-22-13389Receiving quality antenatal care service increases the chance of maternal use of skilled birth attendants in Ethiopia: Using a longitudinal panel surveyPLOS ONE

Dear Dr. Mohammed,

Thank you for submitting your manuscript to PLOS ONE. After careful consideration, we feel that it has merit but does not fully meet PLOS ONE’s publication criteria as it currently stands. Therefore, we invite you to submit a revised version of the manuscript that addresses the points raised during the review process.

 Your manuscript has been assessed by two peer-reviewers and their reports are appended below.  The reviewers comment that your manuscript would benefit from more consistency in the use of terminology across the manuscript. In addition, the reviewers comment that the methodology and data analysis described in this study require additional detail. Furthermore, the discussion section would benefit from a deeper discussion of the implications of this study on future studies and policy making.  Could you please carefully revise the manuscript to address all comments raised?

We look forward to receiving your revised manuscript.

Kind regards,

Maria Elisabeth Johanna Zalm, Ph.D

Editorial Office

PLOS ONE

Journal Requirements:

Reviewers' comments:

Reviewer's Responses to Questions

**Comments to the Author**

1. Is the manuscript technically sound, and do the data support the conclusions?

Reviewer #1: Yes

Reviewer #2: Partly

2. Has the statistical analysis been performed appropriately and rigorously? 

Reviewer #1: I Don't Know

Reviewer #2: Yes

3. Have the authors made all data underlying the findings in their manuscript fully available?

Reviewer #1: Yes

Reviewer #2: Yes

4. Is the manuscript presented in an intelligible fashion and written in standard English?

Reviewer #1: Yes

Reviewer #2: No

5. Review Comments to the Author

Reviewer #1: Dear authors

This is interesting research area. I found the question authors raised very important. Coverage of antenatal care alone couldn’t bring required skilled birth attendance. The coexistence of low quality and coverage ANC in Ethiopia and LMICs, has been obstacle to improve maternal and child health for decades. However, I have also issues to be addressed in the new version of the manuscript. Please, find the comments below:

The authors need be consistent while using terms throughout the manuscript. For instance, postnatal vs postpartum, mothers’ vs women, health care providers vs health professional, skilled birth attendant vs skilled workers, antenatal care vs prenatal care, ANC component quality vs ANC quality etc.

Background

Is quality ANC guarantee to use skilled birth attendant? What is an evidence in high-income countries?

What really contributed to low skilled birth attendant utilization for a women who had at least one ANC visit? Do you think your design is appropriate to explore it?

Method

Study design

“Then, women expected date of delivery was estimated…” how do you overcome recall bias here?

Study population

Who are your source and study population? “All postnatal mothers who had any ANC visit during pregnancy,….” Do you mean all women who received ANC from skilled attendants? What about women who received ANC from other professionals?

Sampling technique

Check how sampling were conducted for panel study? How stratification is performed? “At the second stage, 35 households per cluster were selected and all pregnant women aged 15-49 in the selected households were enrolled and interviewed.”?

Data collection method

“In PMA-Ethiopia panel survey,…” what makes it panel? How many times did they collect the data? This part need detail explanation of tool, data collectors and techniques unless it has been explained elsewhere.

How did you make sure women delivered by skilled attendant (health professional with midwifery skills)? Do you think rural women can differentiate these health professionals? “did you deliver by a skilled attendant?” “type of professional skilled birth attendant”

People rarely measure quality due to its measurement issues; how did you measure quality of ANC? Can we say “quality ANC” by measuring the ANC components only? Have you tried to use model to measure quality? How do you see demand and supply side issues?

Have you considered weighting while calculating ANC quality score from these 13 variables? What will be your answer if reader ask you is it fair to give one for HIV test, taking stool sample, birth preparedness and complication readiness etc.?

Data management and analysis

What were the proportion of missing? How missing data were managed in this analysis?

Why weighting was conducted? How weighting was conducted?

Have checked multi-collinearity? For instance, ANC quality vs ANC4+

“….generalized estimating equation (GEE) Poisson regression model was fitted…” was there significant correlation? Do GEE account for significant correlation within clusters?

Result

Table 1- some category need categorizing, since the cell percentage is very small.

“Number of months pregnant women first talked to health care provider (<=4 months vs >4 months)” what do you mean? Do you mean time of first ANC initiation?

Please take the following to method section; “In this study, 13 major prenatal care services received by mothers were used to create antenatal care quality index from women’s perspective. Women were classified as received ‘better ANC quality’ if they received more than the 75th percentiles of ANC service.”

Type of service received: Have you tried principal component analysis?

No need of repeating the model in the result; “Generalized estimating equation Poisson regression model was used to assess the effect of ANC quality on maternal use of skilled delivery.”

What community level factors or individual level factors did you identify in your case?

Discussion

“Improving the ANC service delivery rather than focusing primarily on facility readiness (inputs) may increase women and partner perceived quality of care because use of services and outcomes are the result not only of the provision of care but also of women’s experience of that care.” what if the low quality ANC was due to inputs/facility readiness? Or what if skilled attendants failed to perform the ANC components due to supply side problems, knowledge, skills, attitude etc.?

What do ANC quality (54%) tell us? Nearly half of the participants received quality ANC….

What is the implication of your study?

Reviewer #2: 1) The abstract needs to be improved. Author should focus more on the findings of this study, and can add important findings in the result section.

2) The use of skilled birth attendant at the time of delivery or following the delivery is not clear in the whole manuscript. Also, author should clarify the role of skilled birth attendants following the delivery. What are the current practices of skilled birth attendant utilization?

3) The rationale of this study is poorly established. Author should explain in the introduction section how skilled birth attendant could prevent maternal deaths. As the study focused that receiving quality antenatal care service increases the chance of maternal use skilled birth attendant at/following the delivery, author should explain how quality antenatal care service utilization can improve the chance of maternal use of skilled birth attendant.

4) Author should mention whether they collect self-reported data or clinically confirmed data. What measures are taken to collect self-reported data?

5) Why author did not consider some important individual level factors like women education or husband education? These factors could have an influence on utilization of skilled birth attendant during or following the delivery.

6) The manuscript has several typo-errors and inconsistency in reference style. Please follow journal guideline and use academic English editing.

6. PLOS authors have the option to publish the peer review history of their article (what does this mean?). If published, this will include your full peer review and any attached files.

Reviewer #1: No

Reviewer #2: **Yes: **Md. Obaidur Rahman

---

## [Author Response · Author response to Decision Letter 0]

22 Nov 2022

Authors response to reviewers comment and suggestion 

Reviewer #1: Dear authors

This is interesting research area. I found the question authors raised very important. Coverage of antenatal care alone couldn’t bring required skilled birth attendance. The coexistence of low quality and coverage ANC in Ethiopia and LMICs, has been obstacle to improve maternal and child health for decades. However, I have also issues to be addressed in the new version of the manuscript. Please, find the comments below:

The authors need be consistent while using terms throughout the manuscript. For instance, postnatal vs postpartum, mothers’ vs women, health care provider’s vs health professional, skilled birth attendant vs skilled workers, antenatal care vs prenatal care, ANC component quality vs ANC quality etc.

Authors: we would like to thank for the valuable suggestions. We have used terms consistently in the revised version. 

Background

Is quality ANC guarantee to use skilled birth attendant? What is an evidence in high-income countries?

Authors: we would also like to thank for raising an important question. As presented in the 2nd and 3rd paragraphs of the introduction section, previous study has shown that mothers who received some of the ANC components have increased their use of skilled birth attendant by 30-50%. In addition, the WHO ANC recommendation report also assured as quality ANC leads women to have safe delivery preference. All these information summary are included in the introduction section.

What really contributed to low skilled birth attendant utilization for a woman who had at least one ANC visit? Do you think your design is appropriate to explore it?

Authors: we also would like to thank for this important comment. Yes, we believe that the design helps in measuring the independent contribution of ANC utilization on utilization of skilled birth attendant by controlling the effect of other factors included in the study, though it is not the strongest design. There may be residual confounders as we have used secondary data. This is considered as a limitation of the study.

Method

Study design

“Then, women expected date of delivery was estimated…” how do you overcome recall bias here?

Authors: thank you again for the comment. In this study, women were enrolled during pregnancy and their expected date of delivery was estimated at the time of enrollment. There may be some recall bias, but will not affect the result, since the expected date of delivery was used for data collection planning purpose. 

Study population

Who are your source and study population? “All postnatal mothers who had any ANC visit during pregnancy,” Do you mean all women who received ANC from skilled attendants? What about women who received ANC from other professionals?

Authors: we would also like to thank for this comment. Yes, the study population were all women who received ANC from skilled attendants. Women who received ANC from other professionals were not included in the source population.

Sampling technique

Check how sampling were conducted for panel study? How stratification is performed? “At the second stage, 35 households per cluster were selected and all pregnant women aged 15-49 in the selected households were enrolled and interviewed.”?

Authors: thank you again for this comment. We have revised the sampling procedure per the comment. Six regions were included in the PMA study and each region were stratified into urban and rural, then enumeration areas (clusters) were selected from both strata using CSA sampling frame. Clusters were selected using probability proportional to size sampling technique. Then, fixed (35) number of households were selected from each cluster and all women in a given household were enrolled and followed. What makes this study panel is that same women enrolled during pregnancy were followed (interviewed) after delivery at 6-8 weeks’ period.

Data collection method

“In PMA-Ethiopia panel survey,…” what makes it panel? How many times did they collect the data? This part need detail explanation of tool, data collectors and techniques unless it has been explained elsewhere.

Authors: we also thank for this comment. What makes this study panel is that the same group of women enrolled during pregnancy were followed and interviewed during postnatal period. In this study, data were collected two times: baseline data at the time of enrolment, and outcome data and other important exposures data after delivery by trained data collectors using interviewer administered questionnaire. This part is revised as per the reviewer’s suggestion.

How did you make sure women delivered by skilled attendant (health professional with midwifery skills)? Do you think rural women can differentiate these health professionals? “did you deliver by a skilled attendant?” “type of professional skilled birth attendant”

Authors: Yes, in rural area many women might not differentiate the specific professional who assisted them. The health professional working at the health facilities are expected to have midwifery skill. For example, women might get assistance from health officer/medical doctor/nurse, who have midwifery level skill in terms of assisting births. The question on type of profession were used to differentiate whether the women were assisted by Health Extension workers ( HEW, those working at health post level only) or traditional births attendants (TBAs), who are not considered as a skilled health workers with the level of midwifery skills. It will not be difficult for women in Ethiopia to know HEWS and TBA, because of they are from the same locality and are well known by the women.

People rarely measure quality due to its measurement issues; how did you measure quality of ANC? Can we say “quality ANC” by measuring the ANC components only? Have you tried to use model to measure quality? How do you see demand and supply side issues?

Authors: we would also like to thank the reviewer for this important comment. Yes, according to the WHO quality framework, ANC has three dimensions: input (service readiness), process (service provision), outcome (client satisfaction). In the current study, we have assessed quality from using service process (provision) quality dimension from the women’s perspective who had already at least 1st ANC visit. This is very important area, but most of the time overlooked by researchers. We haven’t measure quality comprehensively from both demand and supply side. This is one of the limitations of the study. 

Have you considered weighting while calculating ANC quality score from these 13 variables? What will be your answer if reader ask you is it fair to give one for HIV test, taking stool sample, birth preparedness and complication readiness etc.?

Authors: thank you again for this comment. We have considered all the 13 variables having equal importance, since the ministry of health guideline (WHO guideline) recommends each component to be received by a pregnant woman during ANC visit. 

Data management and analysis

What were the proportion of missing? How missing data were managed in this analysis?

Authors: we also thank the reviewer for this comment. We analyzed the complete records without imputation.

Why weighting was conducted? How weighting was conducted?

Authors: we would also like to thank for this comment. we have applied weight to account the non-equal selection probability of individuals within a cluster and improve the sample representativeness and have reliable estimate. Weight in this case is just the inverse of individuals’ selection probability within a cluster.

Have checked multi-collinearity? For instance, ANC quality vs ANC4+

Authors: we would also like to thank for this comment. Yes, we have checked the multi-collinearity between the variables and found no multi-collinearity problem. For example, the corr value for ANC quality and ANC4+ is 0.1821.

“….generalized estimating equation (GEE) Poisson regression model was fitted…” was there significant correlation? Do GEE account for significant correlation within clusters?

Authors: we thank again the reviewer for this important point. We do believe that GEE accounts the within cluster (enumeration area) correlation.

Result

Table 1- some category need categorizing, since the cell percentage is very small.

Authors: we have now merged the cells when possible.

“Number of months’ pregnant women first talked to health care provider (<=4 months vs >4 months)” what do you mean? Do you mean time of first ANC initiation?

Authors: Thank you for the comment. we have revised the statement based on the suggestion. We had used that one as it was stated in the PMA survey questionnaire.

Please take the following to method section; “In this study, 13 major prenatal care services received by mothers were used to create antenatal care quality index from women’s perspective. Women were classified as received ‘better ANC quality’ if they received more than the 75th percentiles of ANC service.”

Authors: Thank you again for this comment. We have moved the statements to the methods section. .

Type of service received: Have you tried principal component analysis?

Authors: We would also like to thank for this important comment. We didn’t apply the PCA in this study. 

No need of repeating the model in the result; “Generalized estimating equation Poisson regression model was used to assess the effect of ANC quality on maternal use of skilled delivery.”

Authors: revised as per the reviewer’s suggestion.

What community level factors or individual level factors did you identify in your case?

Authors: We would also like to thank for this comment. No community level factor considered in the analysis. 

Discussion

“Improving the ANC service delivery rather than focusing primarily on facility readiness (inputs) may increase women and partner perceived quality of care because use of services and outcomes are the result not only of the provision of care but also of women’s experience of that care.” what if the low quality ANC was due to inputs/facility readiness? Or what if skilled attendants failed to perform the ANC components due to supply side problems, knowledge, skills, attitude etc.?

Authors: we would also like to thank for this important comment. As we have explained somewhere above in this response, we haven’t measure quality comprehensively from both demand and supply side. This is one of the limitations of the study and declared in the discussion section. 

What do ANC quality (54%) tell us? Nearly half of the participants received quality ANC….

Authors: We would like to thank for this comment. In this study, ANC quality (39%) tells us that nearly 39% of the women received adequate (more than 75th percentile) recommended ANC services. Sorry, 54% is the proportion of women who had used skilled birth attendant. The calculation is correct, but the mistake was happened when coping to word document from the analysis output tables and we have noticed it after submission to the journal. 

What is the implication of your study?

Authors: we also would like to thank for this comment. This study finding has implication for both practice and further research. The implication for practices is that it will help to plan to improve ANC component quality and as a result improve maternal use of SBA. Also, it has implication for research to qualitatively explore the barriers of quality ANC service provision and other barriers of maternal not use of SBA especially after having skilled ANC visits.

Reviewer #2: 1) The abstract needs to be improved. Author should focus more on the findings of this study, and can add important findings in the result section.

Authors: we would like to thank for this important comment. Now we have added important findings in the result section of the abstract.

2) The use of skilled birth attendant at the time of delivery or following the delivery is not clear in the whole manuscript. Also, author should clarify the role of skilled birth attendants following the delivery. What are the current practices of skilled birth attendant utilization?

Authors: we would also like to thank for this important comment. The role of skilled birth attendants is indicated in the last few sentences of the first paragraph of the background section. That means 3/4th of maternal deaths occurred at the time of delivery or immediately after delivery due to hemorrhage, uterine rupture, sepsis, hypertension, obstructed labor and the like. These causes could have been prevented or managed if women had used Skilled birth attendants.

3) The rationale of this study is poorly established. Author should explain in the introduction section how skilled birth attendant could prevent maternal deaths. As the study focused that receiving quality antenatal care service increases the chance of maternal use skilled birth attendant at/following the delivery, author should explain how quality antenatal care service utilization can improve the chance of maternal use of skilled birth attendant.

Authors: we also would like to thank for this comment. We have explained in the 1st and 2nd paragraph of the background section. The skilled birth attendants prevent maternal death by preventing or managing the major cause of maternal mortality. We have explained the major factors that influence maternal use of Skilled birth attendant, like predisposing factors, enabling factors and service related factors. Here we have questioned ‘why women not use skilled birth attendant or facility delivery while having ANC visit?’ we have explained what study has report about the effect of ANC services on the outcome. In the 2nd paragraph we have shown how receiving most of the ANC services increase women use of skilled birth attendant.

4) Author should mention whether they collect self-reported data or clinically confirmed data. What measures are taken to collect self-reported data?

Authors: we would also like to thank the reviewer for this comment. Data were collected using interviewer administered questionnaire by trained field workers, it is self-reported data. There was intensive training of data collectors and close supervision to increase the reliability of the self-reported data. 

5) Why author did not consider some important individual level factors like women education or husband education? These factors could have an influence on utilization of skilled birth attendant during or following the delivery.

Authors: we would also like to thank for this important comment. As the reviewer rightly mentioned education/literacy is important factor for the current outcome, but we couldn’t consider women education as a potential confounding variable because of it didn’t fulfill confounding variable selection criteria (p-value cutoff <=0.05 or 10% or more difference b/n crude and adjusted effect size estimates). Now, we have also indicated in the methods section how a potential confounding variables were identified and selected, that means using the most commonly used p-value cutoff significance criteria or a 10% or more difference b/n the crude and adjusted effect size estimate. 

6) The manuscript has several typo-errors and inconsistency in reference style. Please follow journal guideline and use academic English editing.

Authors: we also thank for these comments. Now we have corrected the inconsistency in reference style and typo-errors.

---

## [Editor Report · Decision Letter 1]

8 Dec 2022

Receiving quality antenatal care service increases the chance of maternal use of skilled birth attendants in Ethiopia: Using a longitudinal panel survey

PONE-D-22-13389R1

Dear Dr. Mohammed,

We’re pleased to inform you that your manuscript has been judged scientifically suitable for publication and will be formally accepted for publication once it meets all outstanding technical requirements.

Kind regards,

Aklilu Habte Hailegebireal, MPH

Academic Editor

PLOS ONE
---

## [Editor Report · Acceptance letter]

14 Dec 2022

PONE-D-22-13389R1 

Receiving quality antenatal care service increases the chance of maternal use of skilled birth attendants in Ethiopia: Using a longitudinal panel survey 

Dear Dr. Mohammed:

I'm pleased to inform you that your manuscript has been deemed suitable for publication in PLOS ONE. Congratulations! Your manuscript is now with our production department. 

Kind regards, 

on behalf of

Dr. Aklilu Habte Hailegebireal 

Academic Editor

PLOS ONE